# Adversarial Example Games

**Avishek Joey Bose**[*]
Mila, McGill University
joey.bose@mail.mcgill.ca

**Gauthier Gidel**[*]
Mila, Université de Montréal
gauthier.gidel@umontreal.ca

**Hugo Berard**[*]
Mila, Université de Montréal
Facebook AI Research

**Andre Cianflone**
Mila, McGill University

**Pascal Vincent**[†]
Mila, Université de Montréal
Facebook AI Research

**Simon Lacoste-Julien**[†]
Mila, Université de Montréal

**William L. Hamilton**[†]
Mila, McGill University

## Abstract

The existence of adversarial examples capable of fooling trained neural network classifiers calls for a much better understanding of possible attacks to guide the development of safeguards against them. This includes attack methods in the challenging *non-interactive blackbox* setting, where adversarial attacks are generated without any access, including queries, to the target model. Prior attacks in this setting have relied mainly on algorithmic innovations derived from empirical observations (e.g., that momentum helps), lacking principled transferability guarantees. In this work, we provide a theoretical foundation for crafting transferable adversarial examples to entire hypothesis classes. We introduce *Adversarial Example Games* (AEG), a framework that models the crafting of adversarial examples as a min-max game between a generator of attacks and a classifier. AEG provides a new way to design adversarial examples by adversarially training a generator and a classifier from a given hypothesis class (e.g., architecture). We prove that this game has an equilibrium, and that the optimal generator is able to craft adversarial examples that can attack any classifier from the corresponding hypothesis class. We demonstrate the efficacy of AEG on the MNIST and CIFAR-10 datasets, outperforming prior state-of-the-art approaches with an average relative improvement of $29.9\%$ and $47.2\%$ against undefended and robust models (Table 2 & 3) respectively.

## 1 Introduction

Adversarial attacks on deep neural nets expose critical vulnerabilities in traditional machine learning systems [55, 3, 64, 8]. In order to develop models that are robust to such attacks, it is imperative that we improve our theoretical understanding of different attack strategies. While there has been considerable progress in understanding the theoretical underpinnings of adversarial attacks in relatively permissive settings (e.g. *whitebox* adversaries; [53]), there remains a substantial gap between theory and practice in more demanding and realistic threat models.

In this work, we provide a theoretical framework for understanding and analyzing adversarial attacks in the highly-challenging *Non-interactive black**Box** adversary* (NoBox) setting, where the attacker has no direct access, including input-output queries, to the target classifier it seeks to fool. Instead,

---

[*]Equal Contribution, order chosen via randomization.
[†]Canada CIFAR AI Chair

the attacker must generate attacks by optimizing against some representative classifiers, which are assumed to come from a similar hypothesis class as the target.

The NoBox setting is a much more challenging setting than more traditional threat models, yet it is representative of many real-world attack scenarios, where the attacker cannot interact with the target model [15]. Indeed, this setting—as well as the general notion of transferring attacks between classifiers—has generated an increasing amount of empirical interest [25, 51, 73, 71]. The field, however, currently lacks the necessary theoretical foundations to understand the feasibility of such attacks.

**Contributions**. To address this theoretical gap, we cast NoBox attacks as a kind of *adversarial example game* (AEG). In this game, an attacker generates adversarial examples to fool a representative classifier from a given hypothesis class, while the classifier itself is trained to detect the correct labels from the adversarially generated examples. Our first main result shows that the Nash equilibrium of an AEG leads to a distribution of adversarial examples effective against *any* classifier from the given function class. More formally, this adversarial distribution is guaranteed to be the most effective distribution for attacking the hardest-to-fool classifiers within the hypothesis class, providing a worst-case guarantee for attack success against an arbitrary target. We further show that this optimal adversarial distribution admits a natural interpretation as being the distribution that maximizes a form of restricted conditional entropy over the target dataset, and we provide detailed analysis on simple parametric models to illustrate the characteristics of this optimal adversarial distribution. Note that while AEGs are latent games [30], they are distinct from the popular generative adversarial networks (GANs) [32]. In AEGs, there is *no* discrimination task between two datasets (generated one and real one); instead, there is a standard supervised (multi-class) classification task on an adversarial dataset.

Guided by our theoretical results we instantiate AEGs using parametric functions —i.e. neural networks, for both the attack generator and representative classifier and show the game dynamics progressively lead to a stronger attacker and robust classifier pairs. We empirically validate AEG on standard CIFAR and MNIST benchmarks and achieve state-of-the-art performance —compared to existing heuristic approaches— in nearly all experimental settings (e.g., transferring attacks to unseen architectures and attacking robustified models), while also maintaining a firm theoretical grounding.

## 2 Background and Preliminaries

Suppose we are given a classifier $f : \mathcal{X} \to \mathcal{Y}$, an input datapoint $x \in \mathcal{X}$, and a class label $y \in \mathcal{Y}$, where $f(x) = y$. The goal of an adversarial attack is to produce an adversarial example $x' \in \mathcal{X}$, such that $f(x') \neq y$, and where the distance[3] $d(x, x') \leq \epsilon$. Intuitively, the attacker seeks to fool the classifier $f$ into making the wrong prediction on a point $x'$, which is $\epsilon$-close to a real data example $x$.

**Adversarial attacks and optimality**. A popular setting in previous research is to focus on generating *optimal* attacks on a single classifier $f$ [13, 53]. Given a loss function $\ell$, used to evaluate $f$, an adversarial attack is said to be optimal if,

$$x' \in \operatorname{argmax}_{x' \in \mathcal{X}} \ell(f(x'), y), \quad \text{s.t.} \quad d(x, x') \leq \epsilon. \tag{1}$$

In practice, attack strategies that aim to realize (1) optimize adversarial examples $x'$ directly using the gradient of $f$. In this work, however, we consider the more general setting of generating attacks that are optimal against entire hypothesis classes $\mathcal{F}$, a notion that we formalize below.

### 2.1 NoBox Attacks

Threat models specify the formal assumptions of an attack (e.g., the information the attacker is assumed to have access to), which is a core aspect of adversarial attacks. For example, in the popular *whitebox* threat model, the attacker is assumed to have full access to the model $f$'s parameters and outputs [65, 33, 53]. In contrast, the *blackbox* threat model assumes restricted access to the model, e.g., only access to a limited number of input-out queries [18, 42, 57]. Overall, while they consider different access to the target model, traditional whitebox and blackbox attacks both attempt to generate adversarial examples that are optimal for a specific target (i.e., Equation 1).

In this paper, we consider the more challenging setting of **_non_-interactive black_Box_ (_NoBox_)** attacks, intending to generate successful attacks against an unknown target. In the NoBox setting, we assume

no interactive access to a target model; instead, we only assume access to a target dataset and knowledge of the function class to which a target model belongs. Specifically, the NoBox threat model relies on the following key definitions:

- **The target model** $f_t$. The adversarial goal is to attack some target model $f_t : \mathcal{X} \to \mathcal{Y}$, which belongs to an hypothesis class $\mathcal{F}$. Critically, the adversary has *no access* to $f_t$ *at any time*. Thus, in order to attack $f_t$, the adversary must develop attacks that are effective against the entirety of $\mathcal{F}$.

- **The target examples** $\mathcal{D}$. The dataset $\mathcal{D}$ contains the examples $(x, y)$ that attacker seeks to corrupt.

- **An hypothesis class** $\mathcal{F}$. As noted above, we assume that the attacker has access to a hypothesis class $\mathcal{F}$ to which the target model $f_t$ belongs.[4] One can incorporate in $\mathcal{F}$ as much prior knowledge one has on $f_t$ (e.g., the architecture, dataset, training method, or regularization), going from exact knowledge of the target $\mathcal{F} = \{f_t\}$ to almost no knowledge at all (e.g., $\mathcal{F} = \{f \in \text{DenseNets}\}$).

- **A reference dataset** $\mathcal{D}_{\mathbf{ref}}$. The reference dataset $\mathcal{D}_{\text{ref}}$, which is *similar* to the training data of the target model (e.g., sampled from the same distribution) is used to reduce the size of the hypothesis class $\mathcal{F}$ (e.g., we know that the target model perfoms well at classification on $\mathcal{D}_{\text{ref}}$).

- **A representative classifier** $f_c$. Finally, we assume that the attacker has the ability to optimize a representative classifier $f_c$ from the hypothesis class $\mathcal{F}$.

Given these four key components, we formalize the NoBox setting as follows:

**Definition 1.** *The NoBox threat model corresponds to the setting where the attacker (i) knows a hypothesis class $\mathcal{F}$ that the target model $f_t$ belongs to, (ii) has access to a reference dataset $\mathcal{D}_{ref}$ that is similar to the the dataset used to train $f_t$ (e.g., sampled from the same distribution), and (iii) can optimize a representative classifier $f_c \in \mathcal{F}$. The attacker has no other knowledge of—or access to—the target model $f_t$ (e.g., no queries to $f_t$ are allowed). The goal is, for the attacker, to use this limited knowledge to corrupt the examples in a given target dataset $\mathcal{D}$.*

Our definition of a NoBox adversary (Def. 1) formalizes similar notions used in previous work (e.g., see Def. 3 in [67]). Previous work also often refers to related settings as generating *blackbox transfer* attacks, since the goal is to attack the target model $f_t$ while only having access to a representative classifier $f_c$ [25, 51, 73].

Note, that our assumptions regarding dataset access are relatively weak. Like prior work, the attacker is given the target data (i.e., the examples to corrupt) as input, but this is constitutive of the task (i.e., we need access to a target example in order to corrupt it). Our only assumption is to have access to a reference dataset $\mathcal{D}_{\text{ref}}$, which is *similar* to the dataset used to train the target model. We do not assume access to the exact training set. A stronger version of this assumption is made in prior works on blackbox transfer, as these approaches must craft their attacks on a known source model which is pretrained on the same dataset as the target model [67].

## 3   Adversarial Example Games

In order to understand the theoretical feasibility of NoBox attacks, we view the attack generation task as a form of *adversarial game*. The players are the *generator* network $g$—which learns a conditional distribution over adversarial examples—and the representative classifier $f_c$. The goal of the generator network is to learn a conditional distribution of adversarial examples, which can fool the representative classifier $f_c$. The representative classifier $f_c$, on the other hand, is optimized to detect the true label $y$ from the adversarial examples $(x', y)$ generated by $g$. A critical insight in this framework is that the generator and the representative classifier are *jointly* optimized in a maximin game, making the generator's adversarial distribution at the equilibrium theoretically effective against *any* classifier from the hypothesis class $\mathcal{F}$ that $f_c$ is optimized over. At the same time, we will see in Proposition 1 that the min and max in our formulation (AEG) can be switched. It implies that, while optimized, the model $f_c$ converges to a *robust classifier* against any attack generated by the generator $g$ [53, 70], leading to increasingly powerful attacks as the adversarial game progresses.

**Framework**. Given an input-output pair of target datapoints $(x, y) \sim \mathcal{D}$, the generator network $g$ is trained to learn a distribution of adversarial examples $p_{\text{cond}}(\cdot | x, y)$ that—conditioned on an example

to attack $(x, y)$—maps a prior distribution $p_z$ on $\mathcal{Z}$ onto a distribution on $\mathcal{X}$. The classifier network $f_c$ is simultaneously optimized to perform robust classification over the resulting distribution $p_g$ defined in (2) (below). Overall, the generator $g$ and the classifier $f_c$ play the following, two-player zero-sum game:

$$\max_{g \in \mathcal{G}_\epsilon} \min_{f_c \in \mathcal{F}} \mathbb{E}_{(x,y)\sim\mathcal{D}, z\sim p_z} [\ell(f_c(g(x, y, z)), y)] =: \varphi(f_c, g), \qquad \text{(AEG)}$$

where the generator $g \in \mathcal{G}_\epsilon$ is restricted by the similarity constraint $d(g(x, y, z), x) \le \epsilon$, $\forall x, y, z \in \mathcal{X} \times \mathcal{Y} \times \mathcal{Z}$. Once the generator $g$ is trained, one can generate adversarial examples against any classifier in $f_t \in \mathcal{F}$, without queries, by simply sampling $z \sim p_z$ and computing $g(x, y, z)$.

**Connection with NoBox attacks**. The NoBox threat model (Def. 1) corresponds to a setting where the attacker does not know the target model $f_t$ but only a hypothesis class $\mathcal{F}$ such that $f_t \in \mathcal{F}$. With such knowledge, one cannot hope to be better than the *most pessimistic situation* where $f_t$ is the best defender in $\mathcal{F}$. Our maximin formulation (AEG) encapsulates such a worst-case scenario, where the generator aims at finding attacks against the best performing $f$ in $\mathcal{F}$.

**Objective of the generator**. When trying to attack infinite capacity classifiers—i.e., $\mathcal{F}$ contains any measurable function—the goal of the generator can be seen as generating the adversarial distribution $p_g$ with the highest expected conditional entropy $\mathbb{E}_x[\sum_y p_g(y|x) \log p_g(y|x)]$, where $p_g$ is defined as

$$(x', y) \sim p_g \Leftrightarrow x' = g(x, y, z), \ (x, y) \sim \mathcal{D}, \ z \sim p_z \quad \text{with} \quad d(x', x) \le \epsilon. \qquad (2)$$

When trying to attack a specific hypothesis class $\mathcal{F}$ (e.g., a particular CNN architecture), the generator aims at maximizing a notion of restricted entropy defined implicitly through the class $\mathcal{F}$. Thus, the optimal generator in an (AEG) is primarily determined by the statistics of the target dataset $\mathcal{D}$ itself, rather any specifics of a target model. We formalize these high level concepts in §4.2.

**Regularizing the Game**. In practice, the target $f_t$ is usually trained on a non-adversarial dataset and performs well at a standard classification task. In order to reduce the size of the class $\mathcal{F}$, one can bias the representative classifier $f_c$ towards performing well on a standard classification task with respect to $\mathcal{D}_{\text{ref}}$, which leads to the following game:

$$\max_{g \in \mathcal{G}_\epsilon} \min_{f_c \in \mathcal{F}} \mathbb{E}_{(x,y)\sim\mathcal{D}, z\sim p_z}[\ell(f_c(g(x, y, z)), y)] + \lambda \mathbb{E}_{(x,y)\sim\mathcal{D}_{\text{ref}}}[\ell(f_c(x), y)] =: \varphi_\lambda(f, g). \qquad (3)$$

Note that $\lambda = 0$ recovers (AEG). Such modifications in the maximin objective as well as setting the way the models are trained (e.g., optimizer, regularization, additional dataset) biases the training of the $f_c$ and corresponds to an implicit incorporation of prior knowledge on the target $f_t$ in the hypothesis class $\mathcal{F}$. We note that in practice, using a non-zero value for $\lambda$ is essential to achieve the most effective attacks as the prior knowledge acts as a regularizer that incentivizes $g$ to craft attacks against classifiers that behave well on data similar to $\mathcal{D}_{\text{ref}}$.

## 4 Theoretical results

When playing an adversarial example game, the generator and the representative classifier try to beat each other by maximizing their own objective. In games, a standard notion of optimality is the concept of Nash equilibrium [56] where each player cannot improve its objective value by unilaterally changing its strategy. The minimax result in Prop. 1 implies the existence of a Nash equilibrium for the game, consequently providing a well defined target for learning (we want to learn the equilibrium of that game). Moreover, a Nash equilibrium is a stationary point for gradient descent-ascent dynamics; we can thus hope for achieving such a solution by using a gradient-descent-ascent-based learning algorithm on (AEG).[5]

**Proposition 1.** *If $\ell$ is convex (e.g., cross entropy or mean squared loss), the distance $x \mapsto d(x, x')$ is convex for any $x' \in \mathcal{X}$, one has access to any measurable $g$ respecting the proximity constraint in (2), and the hypothesis class $\mathcal{F}$ is convex, then we can switch min and max in (AEG), i.e.,*

$$\min_{f_c \in \mathcal{F}} \max_{g \in \mathcal{G}_\epsilon} \varphi_\lambda(f_c, g) = \max_{g \in \mathcal{G}_\epsilon} \min_{f_c \in \mathcal{F}} \varphi_\lambda(f_c, g) \qquad (4)$$

*Proof sketch.* We first notice that, by (2) any $g$ corresponds to a distribution $p_g$ and thus we have,

$$\varphi(f_c, g) := \mathbb{E}_{(x,y)\sim\mathcal{D},z\sim p_z}[\ell(f_c(g(x,y,z)),y)] = \mathbb{E}_{(x',y)\sim p_g}[\ell(f_c(x'),y)] =: \varphi(f_c, p_g)$$

Consequently, we also have $\varphi_\lambda(f_c, g) = \varphi_\lambda(f_c, p_g)$. By noting $\Delta_\epsilon := \{p_g : g \in \mathcal{G}_\epsilon\}$, we have that,

$$\min_{f_c\in\mathcal{F}} \max_{p_g\in\Delta_\epsilon} \varphi_\lambda(f_c, p_g) = \min_{f_c\in\mathcal{F}} \max_{g\in\mathcal{G}_\epsilon} \varphi_\lambda(f_c, g) \quad \text{and} \quad \max_{p_g\in\Delta_\epsilon} \min_{f_c\in\mathcal{F}} \varphi_\lambda(f_c, p_g) = \max_{g\in\mathcal{G}_\epsilon} \min_{f_c\in\mathcal{F}} \varphi_\lambda(f_c, g)$$

In other words, we can replace the optimization over the generator $g \in \mathcal{G}_\epsilon$ with an optimization over the set of possible adversarial distributions $\Delta_\epsilon$ induced by any $g \in \mathcal{G}_\epsilon$. This equivalence holds by the construction of $\Delta_\epsilon$, which ensures that $\max_{g\in\mathcal{G}_\epsilon} \varphi_\lambda(f_c, g) = \max_{p_g\in\Delta_\epsilon} \varphi_\lambda(f_c, p_g)$ for any $f_c \in \mathbb{F}$.

We finally use Fan's theorem [28] after showing that $(f_c, p_g) \mapsto \varphi_\lambda(f_c, p_g)$ is convex-concave (by convexity of $\ell$ and linearity of $p_g \mapsto \mathbb{E}_{p_g}$) and that $\Delta_\epsilon$ is a compact convex set. In particular, $\Delta_\epsilon$ is compact convex under the assumption that we can achieve any measurable $g$ (detailed in §A). $\qquad\square$

The convexity assumption on the hypothesis class $\mathcal{F}$, Prop. 1 applies in two main cases of interest: (i) infinite capacity, i.e., when $\mathcal{F}$ is any measurable function. (ii) linear classifiers with *fixed* features $\psi : \mathcal{X} \to \mathbb{R}^p$, i.e., $\mathcal{F} = \{w^\top\psi(\cdot), w \in \mathbb{R}^{|\mathcal{Y}|\times p}\}$. This second setting is particularly useful to build intuitions on the properties of (AEG), as we will see in §4.1 and Fig. 1. The assumption that we have access to any measurable $g$, while relatively strong, is standard in the literature and is often stated in prior works as "if $g$ has enough capacity" [33, Prop. 2]. Even if the class of neural networks with a fixed architecture do not verify the assumption of this proposition, the key idea is that neural networks are good candidates to approximate that equilibrium because they are universal approximators [38] and they form a set that is "almost convex" [30]. Proving a similar minimax theorem by only considering neural networks is a challenging problem that has been considered by Gidel et al. [30] in a related setting. It requires a fined grained analysis of the property of certain neural network architecture and is only valid for approximate minimax. We believe such considerations outside of the scope of this work.

## 4.1 A simple setup: binary classification with logistic regression

Let us now consider a binary classification setup where $\mathcal{Y} = \{\pm 1\}$ and $\mathcal{F}$ is the class of linear classifiers with linear features, i.e $f_w(x) = w^\top x$. In this case, the payoff of the game (AEG) is,

$$\varphi(f_\omega, g) := \mathbb{E}_{(x,y)\sim\mathcal{D},\, z\sim p_z}[\log(1 + e^{-y\cdot w^\top g(x,y,z)})] \tag{5}$$

This example is similar to the one presented in [33]. However, our purpose is different since we focus on characterizing the optimal generator in (4). We show that the optimal generator can attack any classifier in $\mathcal{F}$ by shifting the means of the two classes of the dataset $\mathcal{D}$.

**Proposition 2.** *If the generator is allowed to generate any $\ell_\infty$ perturbations. The optimal linear representative classifier is the solution of the following $\ell_1$ regularized logistic regression*

$$w^* \in \arg\min_w \mathbb{E}_{(x,y)\sim\mathcal{D}}[\log(1 + e^{-y\cdot w^\top x + \epsilon\|w\|_1})] . \tag{6}$$

*Moreover if $\omega^*$ has no zero entry, the optimal generator is $g^*(x,y) = x - y \cdot \epsilon\,\mathrm{sign}(w^*)$, is* deterministic *and the pair $(f_{w^*}, g^*)$ is a Nash equilibrium of the game* (5).

A surprising fact is that, unlike in the general setting of Prop. 1, the generator in Prop.2 is deterministic (i.e., does not depend on a latent variable $z$).[6] This follows from the simple structure of classifiers in this class, which allow for a closed form solution for $g^*$. In general, one cannot expect to achieve an equilibrium with a deterministic generator. Indeed, with this example, our goal is simply to illustrate how the optimal generator can attack an entire class of functions with limited capacity: linear classifiers are mostly sensitive to the mean of the distribution of each class; the optimal generator exploits this fact by moving these means closer to the decision boundary.

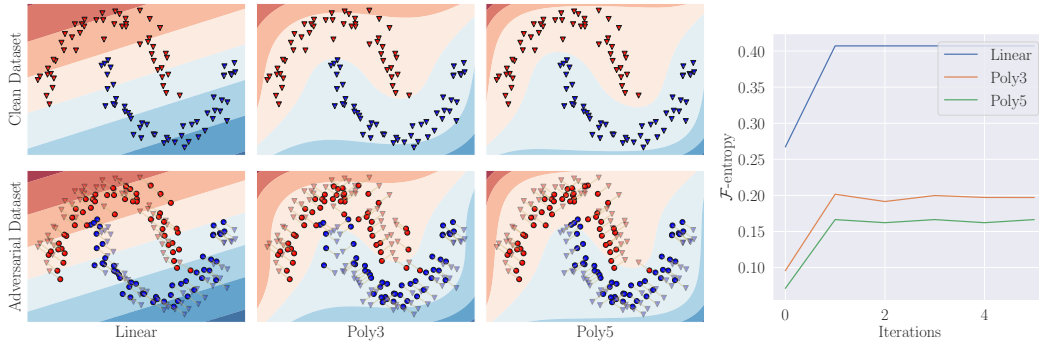

Figure 1: Illustration of Proposition 3 for three classes of classifiers in the context of logistic regression for the two moon dataset of scikit-learn [60] with linear and polynomial (of degree 3 and 5) features. **Left:** Scatter plot of the clean or adversarial dataset and the associated optimal decision boundary. For the adversarial dataset, each corresponding clean example is represented with a ▲/▲ and is connected to its respective adversarial example ●/●. **Right:** value of the $\mathcal{F}$-entropy for the different classes as a function of the number of iterations.

## 4.2 General multi-class classification

In this section, we show that, for a given hypothesis class $\mathcal{F}$, the generated distribution achieving the global maximin against $f_c \in \mathcal{F}$ can be interpreted as the distribution with the highest $\mathcal{F}$-entropy. For a given distribution $p_g$, its $\mathcal{F}$-entropy is the minimum expected risk under $p_g$ one can achieve in $\mathcal{F}$.

**Definition 2.** *For a given distribution $(x, y) \sim p_g$ we define the $\mathcal{F}$-entropy of $p_g$ as*

$$H_{\mathcal{F}}(p_g) := \min_{f_c \in \mathcal{F}} \mathbb{E}_{(x,y) \sim p_g}[\ell(f_c(x), y)] \qquad \text{where } \ell \text{ is the cross entropy loss.} \qquad (7)$$

Thus $\mathcal{F}$-entropy quantifies the amount of "classification information" available in $p_g$ using the class of classifiers $\mathcal{F}$. If the $\mathcal{F}$-entropy is large, $(x, y) \sim p_g$ cannot be easily classified with a function $f_c$ in $\mathcal{F}$. Moreover, it is an upper-bound on the *expected conditional entropy* of the distribution $p_g$.

**Proposition 3.** *The $\mathcal{F}$-entropy is a decreasing function of $\mathcal{F}$, i.e., for any $\mathcal{F}_1 \subset \mathcal{F}_2$,*

$$H_{\mathcal{F}_1}(p_g) \geq H_{\mathcal{F}_2}(p_g) \geq H_y(p_g) := \mathbb{E}_{x \sim p_x}[H(p_g(\cdot|x))].$$

*where $H(p(\cdot|x)) := \sum_{y \in \mathcal{Y}} p(y|x) \ln p(y|x)$ is the entropy of the conditional distribution $p(y|x)$.*

Here $p_g$ is defined as in (2) and implicity depends on $\mathcal{D}$. For a given class $\mathcal{F}$, the solution to an (AEG) game can be seen as one which finds a regularized adversarial distribution of maximal $\mathcal{F}$-entropy,

$$\max_{g \in \mathcal{G}_\epsilon} \min_{f_c \in \mathcal{F}} \varphi_\lambda(f_c, g) = (1 + \lambda) \max_{g \in \mathcal{G}_\epsilon} H_{\mathcal{F}}\left(\frac{1}{(1+\lambda)} p_g + \frac{\lambda}{(1+\lambda)} \mathcal{D}_{\text{ref}}\right)], \qquad (8)$$

where the distribution $\frac{1}{(1+\lambda)} p_g + \frac{\lambda}{(1+\lambda)} \mathcal{D}_{\text{ref}}$ is the mixture of the generated distribution $p_g$ and the empirical distribution over the dataset $\mathcal{D}_{\text{ref}}$. This alternative perspective on the game (AEG) shares similarities with the divergence minimization perspective on GANs [40]. However, while in GANs it represents a divergence between two distributions, in (AEG) this corresponds to a notion of entropy.

A high-level interpretation of $\mathcal{F}$-entropy maximization is that it implicitly defines a metric for distributions which are challenging to classify with only access to classifiers in $\mathcal{F}$. Overall, the optimal generated distribution $p_g$ can be seen as the most adversarial dataset against the class $\mathcal{F}$.

**Properties of the $\mathcal{F}$-entropy**. We illustrate the idea that the optimal generator and the $\mathcal{F}$-entropy depend on the hypothesis class $\mathcal{F}$ using a simple example. To do so, we perform logistic regression (5) with linear and polynomial (of degree 3 and 5) features (respectively called Linear, Poly3, and Poly5) on the two moon dataset of scikit-learn [60]. Note that we have Linear $\subset$ Poly3 $\subset$ Poly5. For simplicity, we consider a deterministic generator $g(x, y)$ that is realized by computing the maximization step via 2D grid-search on the $\epsilon$ neighborhood of $x$. We train our models by successively fully solving the minimization step and the maximization step in (5).

We present the results in Figure 1. One iteration corresponds to the computation of the optimal classifier against the current adversarial distribution $p_g$ (also giving the value of the $\mathcal{F}$-entropy), followed by the computation of the new optimal adversarial $p'_g$ against this new classifier. The left plot illustrates the fact that the way of attacking a dataset depends on the class considered. For instance, when considering linear classifiers, the attack is a uniform translation on all the data-points of the same class. While when considering polynomial features, the optimal adversarial dataset pushes the the corners of the two moons closer together. In the right plot, we can see an illustration of Proposition 3, where the $\mathcal{F}$-entropy takes on a smaller value for larger classes of classifiers.

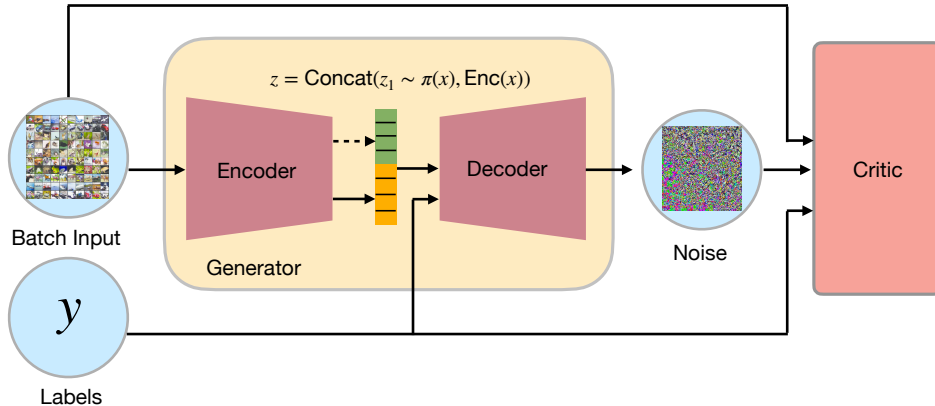

Figure 2: AEG framework architecture

## 5 Attacking in the Wild: Experiments and Results

We investigate the application of our AEG framework to produce adversarial examples against MNIST and CIFAR-10 classifiers. First we investigate our performance in a challenging NoBox setting where we must attack an unseen target model with knowledge of only its hypothesis class (i.e., architecture) and a sample of similar training data (§5.1). Following this, we investigate how well AEG attacks transfer across architectures (§5.2), as well as AEG's performance attacking robust classifiers (§5.3).

**Experimental setup**. We perform all attacks, including baselines, with respect to the $\ell_\infty$ norm constraint with $\epsilon = 0.3$ for MNIST and $\epsilon = 0.03125$ for CIFAR-10. For AEG models, we train both generator ($g$) and representative classifier ($f_c$) using stochastic gradient descent-ascent with the ExtraAdam optimizer [29] and held out target models, $f_t$, are trained offline using SGD with Armijo line search [69]. Full details of our model architectures, including hyperparameters, employed in our AEG framework can be found in Appendix §D.[7]

**Baselines**. Throughout our experiments we rely on four standard blackbox transfer attack strategies adapted to the NoBox setting: the Momentum-Iterative Attack (MI-Attack) [24], the Input Diversity (DI-Attack) [73], the Translation-Invariant (TID-Attack) [25] and the Skip Gradient Method (SGM-Attack) [71]. For fair comparison, we inherit all hyperparameter settings from their respective papers. Note that SGM-attack is only defined with architectures that contain skip connections (e.g. ResNets).

**AEG Architecture**. The high-level architecture of our AEG framework is illustrated in Figure 2. The generator takes the input $x$ and encode it into $\psi(x)$, then the generator uses this encoding to compute a probability vector $p(\psi(x))$ in the probability simplex of size $K$, the number of classes. Using this probability vector, the network then samples a categorical variable $z$ according to a multinomial distribution of parameter $p(\psi(x))$. Intuitively, this category may correspond to a target for the attack. The gradient is backprogated across this categorical variable using the gumble-softmax trick [45, 52]. Finally, the decoder takes as input $\psi(x)$, $z$ and the label $y$ to output an adversarial perturbation $\delta$ such that $\|\delta\| \leq \epsilon$. In order to generate adversarial perturbations over images that obey $\epsilon$-ball constraints, we employ a scaled tanh output layer to scale the output of the generator to $(0, 1)$, subtract the clean images, and finally apply an elementwise multiplication by $\epsilon$. We then compute $\ell(f(x + \delta), y)$ where $f$ is the critic and $\ell$ the cross entropy loss. Further details can be found in Appendix §D.

### 5.1 NoBox Attacks on a Known Architecture Class but Unknown Train Set

We first evaluate the AEG framework in a NoBox setting, where we know only the architecture of the target model and have only access to a sample of similar training data (but not the exact training data of the target model). To simulate having access to a similar (but not identical dataset) as the target model, for each dataset we create random equally-sized splits of the data (10000 examples per splits). Within each split we use one fold to train the split classifier which acts as the representative classifiers for all attackers who are then evaluated their ability to fool the remaining split classifiers

on unseen target examples $\mathcal{D}$. For the MNIST dataset we consider LeNet classifier [47], while for CIFAR-10 we consider ResNet-18 [37]. Table 5.1 shows the results of our experiments on this task, averaged across all splits and folds. We see that our AEG approach achieves state-of-the-art results, either outperforming or matching (within a 95% confidence interval) all baselines in both settings. Note that this task is significantly more challenging than many prior blackbox attack setups, which assume access to the full training data of the target model.[8]

| Dataset | MI-Attack | DI-Attack | TID-Attack | SGM-Attack | AEG (Ours) |
|---------|-----------|-----------|------------|------------|------------|
| MNIST | $87.5 \pm 2.7$ | $\mathbf{89.5 \pm 2.5}$ | $85.4 \pm 2.8$ [†] | N/A | $\mathbf{89.5 \pm 3.2}$ |
| CIFAR-10 (Res18) | $56.8 \pm 1.2$ [†] | $84.0 \pm 1.5$ [†] | $9.1 \pm 1.6$ [†] | $60.5 \pm 1.5$ [†] | $\mathbf{87.0 \pm 2.1}$ |

Table 1: Attack success rates, averaged across target models with 95% confidence intervals shown. [†]indicates a statistically significant result as determined by the paired T-test when compared to AEG.

## 5.2 NoBox Attacks Across Distinct Architectures

We now consider NoBox attacks where we do not know the architecture of the target model but where the training data is known—a setting previously referred to as blackbox transfer [67]. For evaluation, we use CIFAR-10 and train 10 instances of VGG-16 [62], ResNet-18 (RN-18) [37], Wide ResNet (WR) [75], DenseNet-121 (DN-121) [39] and Inception-V3 architectures (Inc-V3) [66]. Here, we optimize the attack approaches against a single pre-trained classifier from a particular architecture and then evaluate their attack success on classifiers from distinct architectures averaged over 5 instantiations. Our findings when using ResNet-18, DenseNet-121 and the VGG-16 as the source architecture are provided in Table 2. Overall we find that AEG beats all other approaches and lead to a new state of the art. In particular AEG outperforms the best baseline in each setting by an average of 29.9% across the different source architectures with individual average gains of 9.4%, 36.2%, and 44.0% when using a RN-18 model, DN-121, and VGG-16 source models respectively.

| Source | Attack | VGG-16 | RN-18 | WR | DN-121 | Inc-V3 |
|--------|--------|--------|-------|-----|--------|--------|
| | Clean | $11.2 \pm 1.8$ | $13.1 \pm 4.0$ | $6.8 \pm 1.4$ | $11.2 \pm 2.8$ | $9.9 \pm 2.6$ |
| RN-18 | MI-Attack | $63.9 \pm 2.6$ | $74.6 \pm 0.8$ | $63.1 \pm 2.4$ | $72.5 \pm 2.6$ | $67.9 \pm 3.2$ |
| | DI-Attack | $77.4 \pm 3.4$ | $90.2 \pm 1.6$ | $74.0 \pm 2.0$ | $87.1 \pm 2.6$ | $\mathbf{85.8 \pm 1.6}$ |
| | TID-Attack | $21.6 \pm 2.6$ | $26.5 \pm 4.8$ | $14.0 \pm 3.0$ | $22.3 \pm 3.2$ | $19.8 \pm 1.8$ |
| | SGM-Attack | $68.4 \pm 3.6$ | $79.5 \pm 1.0$ | $64.3 \pm 3.2$ | $73.8 \pm 2.0$ | $70.6 \pm 3.4$ |
| | AEG (Ours) | $\mathbf{93.8 \pm 0.7}$ | $\mathbf{97.1 \pm 0.4}$ | $\mathbf{80.2 \pm 2.2}$ | $\mathbf{93.1 \pm 1.3}$ | $\mathbf{88.4 \pm 1.6}$ |
| DN-121 | MI-Attack | $54.3 \pm 2.2$ | $62.5 \pm 1.8$ | $56.3 \pm 2.6$ | $66.1 \pm 3.0$ | $65.0 \pm 2.6$ |
| | DI-Attack | $61.1 \pm 3.8$ | $69.1 \pm 1.6$ | $61.9 \pm 2.2$ | $77.1 \pm 2.4$ | $71.6 \pm 3.2$ |
| | TID-Attack | $21.7 \pm 2.4$ | $23.8 \pm 3.0$ | $14.0 \pm 2.8$ | $21.7 \pm 2.2$ | $19.3 \pm 2.4$ |
| | SGM-Attack | $51.6 \pm 1.4$ | $60.2 \pm 2.6$ | $52.6 \pm 1.8$ | $64.7 \pm 3.2$ | $61.4 \pm 2.6$ |
| | AEG (Ours) | $\mathbf{93.7 \pm 1.0}$ | $\mathbf{97.3 \pm 0.6}$ | $\mathbf{81.8 \pm 3.0}$ | $\mathbf{96.7 \pm 0.8}$ | $\mathbf{92.7 \pm 1.6}$ |
| VGG-16 | MI-Attack | $49.9 \pm 0.2$ | $50.0 \pm 0.4$ | $46.7 \pm 0.8$ | $50.4 \pm 1.2$ | $50.0 \pm 0.6$ |
| | DI-Attack | $65.1 \pm 0.2$ | $64.5 \pm 0.4$ | $58.8 \pm 1.2$ | $64.1 \pm 0.6$ | $60.9 \pm 1.2$ |
| | TID-Attack | $26.2 \pm 1.2$ | $24.0 \pm 1.2$ | $13.0 \pm 0.4$ | $20.8 \pm 1.4$ | $18.8 \pm 0.4$ |
| | AEG (Ours) | $\mathbf{97.5 \pm 0.4}$ | $\mathbf{96.1 \pm 0.5}$ | $\mathbf{85.2 \pm 2.2}$ | $\mathbf{94.1 \pm 1.2}$ | $\mathbf{89.5 \pm 1.3}$ |

Table 2: Error rates on $\mathcal{D}$ for average NoBox architecture transfer attacks with $\epsilon = 0.03125$. The $\pm$ correspond to 2 standard deviations (95.5% confidence interval for normal distributions).

## 5.3 NoBox Attacks Against Robust Classifiers

We now test the ability of our AEG framework to attack target models that have been robustified using adversarial and ensemble adversarial training [53, 67]. For evaluation against PGD adversarial training, we use the public models as part of the MNIST and CIFAR-10 adversarial examples

challenge.[9] For ensemble adversarial training, we follow the approach of Tramèr et al. [67] (see Appendix D.3). We report our results in Table 3 and average the result of stochastic attacks over 5 runs. We find that AEG achieves state-of-the-art performance in all settings, proving an average improvement in success rates of $54.1\%$ across all robustified MNIST models and $40.3\%$ on robustified CIFAR-10 models.

| Dataset | Defence | Clean | MI-Att[†] | DI-Att | TID-Att | SGM-Att[†] | AEG (Ours) |
|---------|---------|-------|--------|--------|---------|---------|------------|
| MNIST | $A_{ens4}$ | 0.8 | 43.4 | 42.7 | 16.0 | N/A | **65.0** |
| | $B_{ens4}$ | 0.7 | 20.7 | 22.8 | 8.5 | N/A | **50.0** |
| | $C_{ens4}$ | 0.8 | 73.8 | 30.0 | 9.5 | N/A | **80.0** |
| | $D_{ens4}$ | 1.8 | 84.4 | 76.0 | 81.3 | N/A | **86.7** |
| | Madry-Adv | 0.8 | 2.0 | 3.1 | 2.5 | N/A | **5.9** |
| CIFAR-10 | $RN\text{-}18_{ens3}$ | 16.8 | 17.6 | 21.6 | 33.1 | 19.9 | **52.2** |
| | $WR_{ens3}$ | 12.8 | 18.4 | 20.6 | 28.8 | 18.0 | **49.9** |
| | $DN\text{-}121_{ens3}$ | 21.5 | 20.3 | 22.7 | 31.3 | 21.9 | **41.4** |
| | $Inc\text{-}V3_{ens3}$ | 14.8 | 19.5 | 42.2* | 30.2* | 35.5* | **47.5** |
| | Madry-Adv | 12.9 | 17.2 | 16.6 | 16.6 | 16.0 | **21.6** |

Table 3: Error rates on $\mathcal{D}$ for NoBox known architecture attacks against Adversarial Training and Ensemble Adversarial Training. * Attacks were done using WR. [†] Deterministic attack.

# 6   Related Work

In addition to non-interactive blackbox adversaries we compare against, there exists multiple hybrid approaches that combine crafting attacks on surrogate models which then serve as a good initialization point for queries to the target model [57, 61, 41]. Other notable approaches to craft blackbox transfer attacks learning ghost networks [48], transforming whitebox gradients with small ResNets [50], and transferability properties of linear classifiers and 2-layer ReLu Networks [17]. There is also a burgeoning literature of using parametric models to craft adversarial attacks such as the Adversarial Transformation Networks framework and its variants [4, 72]. Similar in spirit to our approach many attacks strategies benefit from employing a latent space to craft attacks [76, 68, 9]. However, unlike our work, these strategies cannot be used to attack entire hypothesis classes.

Adversarial prediction games between a learner and a data generator have also been studied in the literature [11], and in certain situations correspond to a Stackelberg game Brückner and Scheffer [10]. While similar in spirit, our theoretical framework is tailored towards crafting adversarial attacks against a fixed held out target model in the novel NoBox threat model and is a fundamentally different attack paradigm. Finally, Erraqabi et al. [27] also investigate an adversarial game framework as a means for building robust representations in which an additional discriminator is trained to discriminate adversarial example from natural ones, based on the representation of the current classifier.

# 7   Conclusion

In this paper, we introduce the Adversarial Example Games (AEG) framework which provides a principled foundation for crafting adversarial attacks in the NoBox threat model. Our work sheds light on the existence of adversarial examples as a natural consequence of restricted entropy maximization under a hypothesis class and leads to an actionable strategy for attacking all functions taken from this class. Empirically, we observe that our approach leads to state-of-the-art results when generating attacks on MNIST and CIFAR-10 in a number of challenging NoBox attack settings. Our framework and results point to a promising new direction for theoretically-motivated adversarial frameworks. However, one major challenge is scaling up the AEG framework to larger datasets (e.g., ImageNet), which would involve addressing some of the inherent challenges of saddle point optimization [5]. Investigating the utility of the AEG framework for training robustified models is another natural direction for future work.

## Broader Impact

Adversarial attacks, especially ones under more realistic threat models, pose several important security, ethical, and privacy risks. In this work, we introduce the NoBox attack setting, which generalizes many other blackbox transfer settings, and we provide a novel framework to ground and study attacks theoretically and their transferability to other functions within a class of functions. As the NoBox threat model represents a more realistic setting for adversarial attacks, our research has the potential to be used against a class of machine learning models in the wild. In particular, in terms of risk, malicious actors could use approaches based on our framework to generate attack vectors that compromise production ML systems or potentially bias them toward specific outcomes.

As a concrete example, one can consider creating transferrable examples in the physical world, such as the computer vision systems of autonomous cars. While prior works have shown the possibility of such adversarial examples —i.e., adversarial traffic signs, we note that there is a significant gap in translating synthetic adversarial examples to adversarial examples that reside in the physical world [45]. Understanding and analyzing the NoBox transferability of adversarial examples to the physical world—in order to provide public and academic visibility on these risks—is an important direction for future research. Based on the known risks of designing new kinds of adversarial attacks—discussed above—we now outline the ways in which our research is informed by the intent to mitigate these potential societal risks. For instance, our research demonstrates that one can successfully craft adversarial attacks even in the challenging NoBox setting. It raises many important considerations when developing robustness approaches. A straightforward extension is to consider our adversarial example game (AEG) framework as a tool for training robust models. On the theoretical side, exploring formal verification of neural networks against NoBox adversaries is an exciting direction for continued exploration. As an application, ML practitioners in the industry may choose to employ new forms of A/B testing with different types of adversarial examples, of which AEG is one method to robustify and stress test production systems further. Such an application falls in line with other general approaches to red teaming AI systems [10] and verifiability in AI development. In essence, the goal of such approaches, including adversarial examples for robustness, is to align AI systems' failure modes to those found in human decision making.

## Acknowledgments and Disclosure of Funding

The authors would like to acknowledge Olivier Mastropietro, Chongli Qin and David Balduzzi for helpful discussions as well as Sebastian Lachapelle, Pouya Bashivan, Yanshuai Cao, Gavin Ding, Ioannis Mitliagkas, Nadeem Ward, and Damien Scieur for reviewing early drafts of this work.

**Funding.** This work is partially supported by the Canada CIFAR AI Chair Program (held at Mila), NSERC Discovery Grant RGPIN-2019-05123 (held by Will Hamilton at McGill), NSERC Discovery Grant RGPIN-2017-06936, an IVADO Fundamental Research Project grant PRF-2019-3583139727, and a Google Focused Research award (both held at U. Montreal by Simon Lacoste-Julien). Joey Bose was also supported by an IVADO PhD fellowship, Gauthier Gidel by a Borealis AI fellowship and by the Canada Excellence Research Chair in "Data Science for Real-Time Decision-making" (held at Polytechnique by Andrea Lodi), and Andre Cianflone by a NSERC scholarship and a Borealis AI fellowship. Simon Lacoste-Julien and Pascal Vincent are CIFAR Associate Fellows in the Learning in Machines & Brains program. Finally, we thank Facebook for access to computational resources.

**Competing interests.** Joey Bose was formerly at FaceShield.ai which was acquired in 2020. W.L. Hamilton was formerly a Visiting Researcher at Facebook AI Research. Simon Lacoste-Julien additionally works part time as the head of the SAIT AI Lab, Montreal from Samsung.

## Footnotes

[3]We assume that the $\ell_\infty$ is used in this work, [33, 53] , but our results generalize to any distance $d$.

[4]Previous work [67] usually assumes to have access to the architecture of $f_t$; we are more general by assuming access to a hypothesis class $\mathcal{F}$ containing $f_t$; e.g., DenseNets can represent ConvNets.

[5]Note that, similarly as in practical GANs training, when the classifier and the generator are parametrized by neural networks, providing convergence guarantees for a gradient based method in such a nonconvex-nonconcave minimax game is an open question that is outside of the scope of this work.

[6]Note also that one can generalize Prop. 2 to a perturbation with respect to a general norm $\|\cdot\|$, in that case, the $\epsilon$-regularization for the classifier would be with respect to the dual norm $\|\cdot\|_* := \max_{\|u\|\leq 1}\langle\cdot, u\rangle$. E.g., as previously noted by Goodfellow et al. [33], $\ell_\infty$ adversarial perturbation leads to a $\ell_1$ regularization.

[7]Code: https://github.com/joeybose/Adversarial-Example-Games.git

[8]We include results on a more permissive settings with access to the full training data in Appendix C.1

[9] https://github.com/MadryLab/[x]_challenge, for [x] in {cifar10, mnist}. Note that our threat model is more challenging than these challenges as we use non-robust source models.

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
