[Supplementary Material]

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

Let us consider the case where $\mathcal{X}$ is finite as a warm-up. In practice, this can be the case if we consider that for instance one only allows a finite number of values for the pixels, e.g. (integers between 0 and 255 for CIFAR-10). In that case, we have that

$$\mathbb{P}_{adv}(x, y) = \sum_{x' \in \mathcal{X}} \mathbb{P}_{data}(x', y) \mathbb{P}_g(x|x', y) \quad \text{where} \quad d(x, x') > \epsilon \Rightarrow \mathbb{P}_g(x|x', y) = 0 \,. \tag{10}$$

Assuming that one can achieve any $\mathbb{P}_g(\cdot|x', y)$ respecting the proximity constraint, the set $\{\mathbb{P}_{adv}\}$ is convex and compact. It is compact because closed and bounded in finite dimension and convex because of the linear dependence in $\mathbb{P}_g$ in (10) (and the fact that if $\mathbb{P}_{g_1}$ and $\mathbb{P}_{g_2}$ respect the constraints then $\lambda \mathbb{P}_{g_1} + (1 - \lambda)\mathbb{P}_{g_2}$ does it too).

For the non finite input case, we can consider that the generator is a random variable defined on the probability space $(\mathcal{X} \times \mathcal{Y} \times \mathcal{Z}, \mathcal{B}, \mathbb{P}_{(x,y)} \times \mathbb{P}_z := \mathbb{P})$ where $\mathcal{B}$ is the Borel $\sigma$-algebra, $\mathbb{P}_{(x,y)}$ the probability on the space of data and $\mathbb{P}_z$ the probability on the latent space. Then the adversarial distributions $p_g$ we consider is the pushforward distributions $\mathbb{P} \circ G^{-1}$ with $G$ such that,

$$G(x, y, z) = (g(x, y, z), y) \quad \text{and} \quad d(g(x, y, z), x) \le \epsilon, \forall x, y, z \in \mathcal{X} \times \mathcal{Y} \times \mathcal{Z} \,. \tag{11}$$

It implies that $\Delta_\epsilon := \{p_g \,:\, g \in \mathcal{G}_\epsilon\} = \{\mathbb{P} \circ G^{-1} \,|\, G \text{ measurable satisfying (11)}\}$. By definition of the payoff $\varphi$ and (2), we have that

$$\varphi(f_c, g) := \mathbb{E}_{(x,y) \sim \mathcal{D}, z \sim p_z}[\ell(f_c(g(x, y, z)), y)] = \mathbb{E}_{(x',y) \sim p_g}[\ell(f_c(x'), y)] =: \varphi(f_c, p_g)$$

and thus it lead to the equivalence,

$$\min_{f_c \in \mathcal{F}} \max_{p_g \in \Delta_\epsilon} \varphi_\lambda(f_c, p_g) = \max_{p_g \in \Delta_\epsilon} \min_{f_c \in \mathcal{F}} \varphi_\lambda(f_c, p_g) \iff \min_{f_c \in \mathcal{F}} \max_{g \in \mathcal{G}_\epsilon} \varphi_\lambda(f_c, g) = \max_{g \in \mathcal{G}_\epsilon} \min_{f_c \in \mathcal{F}} \varphi_\lambda(f_c, g)$$

Recall that we assumed that one has access to any measurable $g$ that satisfies (11), the minimax problem (4). Let us show that under this assumption the set $\Delta_\epsilon$ is convex and compact.

$\Delta_\epsilon$ is convex: let us consider $\mathbb{P} \circ G_1^{-1}$ and $\mathbb{P} \circ G_2^{-1}$ we have that

$$\lambda \mathbb{P} \circ G_1^{-1} + (1 - \lambda)\mathbb{P} \circ G_2^{-1} = \mathbb{P} \circ G_3^{-1} \tag{12}$$

where $g_3(x, y, z) = \delta(z)g_1(x, y, z) + (1 - \delta(z))g_2(x, y, z)$ and where $\delta \sim Ber(\lambda)$. Note that $g_3$ satisfies (11) by convexity of $x \mapsto d(x, x')$.

$\Delta_\epsilon$ is compact: By using Skorokhod's representation theorem [7] we can show that this set is closed and thus compact (as closed subsets of compact sets are compact) using the weak convergence of measures as topology.

Thus $\Delta_\epsilon$ is a convex compact Haussdorf space and in both cases ($\mathcal{X}$ finite and infinite) and we can apply Fan's Theorem.

**Theorem 1.** *[28, Theorem 2] Let $U$ be a compact and convex Hausdorff space and $V$ an arbitrary convex set. Let $\varphi$ be a real valued function on $U \times V$ such that for every $v \in V$ the function $\varphi(\cdot, v)$ is lower semi-continuous on $U$. If $\varphi$ is convex-concave then,*

$$\min_{u \in U} \sup_{v \in V} \varphi(u, v) = \sup_{v \in V} \min_{u \in U} \varphi(u, v) \tag{13}$$

Note that we do not prove this result for neural networks. □

**Proposition 2.** *If the generator is allowed to generate any $\ell_\infty$ perturbations. The optimal linear representative classifier is the solution of the following $\ell_1$ regularized logistic regression*

$$w^* \in \arg\min_w \mathbb{E}_{(x,y)\sim\mathcal{D}}[\log(1 + e^{-y \cdot w^\top x + \epsilon\|w\|_1})]\,. \tag{6}$$

*Moreover if $\omega^*$ has no zero entry, the optimal generator is $g^*(x,y) = x - y \cdot \epsilon \operatorname{sign}(w^*)$, is* deterministic *and the pair $(f_{w^*}, g^*)$ is a Nash equilibrium of the game* (5).

*Proof.* We prove the result here for any given norm $\|\cdot\|$. Let us consider the loss for a given pair $(x, y)$

$$\log(1 + e^{y(w^\top g(x,y)+b)}) \tag{14}$$

then by the fact that $x \mapsto \log(1 + e^x)$ is increasing, maximizing this term for $\|g(x,y) - x\|_\infty \leq \epsilon$, boils down to solving the following maximization step,

$$\max_{\delta,\,\|\delta\|\leq\epsilon} y(w^\top \delta) = \|w\|_* \tag{15}$$

Particularly, for the $\ell_\infty$ norm we get

$$\arg\max_{\delta,\,\|\delta\|_\infty\leq\epsilon} y(w^\top \delta) = \epsilon y \operatorname{sign}(w)\,. \tag{16}$$

and

$$\max_g \mathbb{E}_{(x,y)\sim p_{data}}[\log(1 + e^{y(w^\top g(x,y)+b)})] = \mathbb{E}_{(x,y)\sim p_{data}}[\log(1 + e^{y(w^\top x+b)+\epsilon\|w\|_1})] \tag{17}$$

To show that $(f^*, g^*)$ is a Nash equilibrium of the game (5), we first notice that by construction

$$\varphi(f^*, g^*) = \min_{f\in\mathcal{F}} \max_g \varphi(f, g) \tag{18}$$

where $\mathcal{F}$ is the class of classifier with linear logits. We then just need to notice that for all $f \in \mathcal{F}$ we have,

$$\varphi(f, g^*) = \mathbb{E}_{(x,y)\sim p_{data}}[\log(1 + e^{-y(w^\top x+b)+\epsilon w^\top \operatorname{sign}(w^*)})] \tag{19}$$

That is a convex problem in $(w, b)$. Since, in a neighborhood of $w^*$ we have that $w^\top \operatorname{sign}(w^*) = \|w\big|_{\operatorname{supp}(w^*)}\|_1$. Thus by assuming that $w^*$ is full support and since $w^*$ minimize (6) we have that

$$\nabla_w \varphi(w^*, b^*, g^*) = \nabla_w \mathbb{E}_{(x,y)\sim p_{data}}[\log(1 + e^{-y(w^\top x+b)+\epsilon\|w\|_1})] = 0 \tag{20}$$

Finally, by convexity of the problem (19) we can conclude that $w^*$ is a minimizer of $\varphi(\cdot, g^*)$. To sum-up we have that

$$\varphi(f^*, g) \leq \varphi(f^*, g^*) \leq \varphi(f, g^*) \tag{21}$$

meaning that $(f^*, g^*)$ is a Nash equilibrium of (5). □

**Proposition 3.** *The $\mathcal{F}$-entropy is a decreasing function of $\mathcal{F}$, i.e., for any $\mathcal{F}_1 \subset \mathcal{F}_2$,*

$$H_{\mathcal{F}_1}(p_g) \geq H_{\mathcal{F}_2}(p_g) \geq H_y(p_g) := \mathbb{E}_{x\sim p_x}[H(p_g(\cdot|x))]\,.$$

*where $H(p(\cdot|x)) := \sum_{y\in\mathcal{Y}} p(y|x)\ln p(y|x)$ is the entropy of the conditional distribution $p(y|x)$.*

*Proof.* In the case where $f^*(x) := p(y|x) \in \mathcal{F}$, since $f^*$ a minimizer of the expected cross entropy loss over the class of any function, we have that

$$H_y(p) := \min_{f\in\mathcal{X}^{\mathcal{Y}}} \mathbb{E}_{(x,y)\sim p}[\ell(f(x), y)] = \mathbb{E}_x[H(p(\cdot|x))] \tag{22}$$

**Lemma 1.** *Given a data distribution $(x, y) \sim p_{adv}$ a minimizer of the cross entropy loss is*

$$p_{adv}(y|\cdot) \in \arg\min_f \mathbb{E}_{(x,y)\sim p_{adv}}[\ell(f(x), y)]\,. \tag{23}$$

*Proof.* Let us start by noticing that,

$$\min_f \mathbb{E}_{(x,y)\sim p}[\ell(f(x),y)] = \mathbb{E}_{x\sim p_x} \min_{q=f(x)} \mathbb{E}_{y\sim p(\cdot|x)}[\ell(q,y)] \tag{24}$$

Using the fact that $\ell$ is the cross-entropy loss we get

$$\mathbb{E}_{y\sim p(\cdot|x)}[\ell(q,y)] = \mathbb{E}_{y\sim p(\cdot|x)}[-\sum_{i=1}^{K} y_i \ln(q_i)] = -\sum_{i=1}^{K} p_i \ln(q_i) \tag{25}$$

where we noted $p_i = p(y = i|x)$. Noticing that since $q_i$ is a probability distribution we have $\sum_{i=1}^{K} q_i = 1$, we have,

$$\mathbb{E}_{y\sim p(\cdot|x)}[\ell(q,y)] = -\sum_{i=1}^{K-1} p_i \ln(q_i) - p_K \ln(1 - \sum_{i=1}^{K-1} q_i) \tag{26}$$

we can then differentiate this loss with respect to $q_i \geq 0$ and get,

$$\frac{\partial \mathbb{E}_{y\sim p(\cdot|x)}[\ell(q,y)]}{\partial q_i}(q) = -\frac{p_i}{q_i} + \frac{p_K}{q_K} \tag{27}$$

We can finally notice that $q_i = p_i$ is a feasible solution. ☐

☐

# B    Experimental Details

The experiments are subject to different sources of variations, in all our experiments we try to take into account those sources of variations when reporting the results. We detail the different sources of variations for each experiment and how we report them in the next section.

## B.1    Source of variations

**NoBox attacks on a known architecture class**   we created a random split of the MNIST and CIFAR10 dataset and trained a classifier on each splits. To evaluate each method we then use one of the classifiers as the source classifier and all the other classifiers as the targets we want to attack. We then compute the mean and standard deviation of the attack success rates across all target classifiers. To take into account the variability in the results that comes from using a specific classifier as the source model, we also repeat the evaluation by changing the source model. We report the average and $95\%$ interval (assuming the results follow a normal distribution) in Table 5.1 by doing macro-averaging overall evaluations.

**NoBox attacks across distinct architectures**   For each architecture, we trained 10 different models. When evaluated against a specific architecture we evaluate against all models of this architecture. In Table 2, we report the mean and standard deviation of the error rates across all models.

**NoBox Attacks against robust classifiers**   For this experiment, we could only train a single robust target model per architecture because of our computational budget. The only source of variations is thus due to the inherent stochasticity of each method. Evaluating this source of randomness would require to run each method several times, unfortunately, this is quite expensive and our computational budget didn't allow for it. In Table 3, we thus only report a single number per architecture.

# C    Additional results

## C.1    Quantitative Results

We now provide additional results in the form of whitebox and blackbox query attacks adapted to the NoBox evaluation protocol for Known-Architecture attacks which is the experimental setting in

**Q1**. For whitebox attacks we evaluate APGD-CE and APGD-DLR [20] which are improvements over the powerful PGD attack [53]. When ensembled with another powerful perturbation minimizing whitebox attack FAB [19] and the query efficient blackbox Square attacks [2] yields the current SOTA attack strategy called AutoAttack [20] [20]. Additionally, we compare with two parametric blackbox query approaches that both utilize a latent space in AutoZoom [68] and $\mathcal{N}$ Attack [49]. To test transferability of whitebox and blackbox query attacks in the NoBox known architecture setting we give generous iteration and query budgets (10x the reported settings in the original papers) when attacking the source models, but only a single query for each target model. It is interesting to note that APGD variant whitebox attacks are significantly more effective than query based blackbox attacks but lack the same effectiveness of NoBox baselines. We hypothesize that the transferability of whitebox attacks may be due to the fact that different functions learn similar decision boundaries but different enough such that minimum distortion whitebox attacks such as FAB are ineffective.

| | | MNIST | CIFAR-10 |
|---|---|---|---|
| | AutoAttack* | $84.4 \pm 5.1$ | $91.0 \pm 1.9$ |
| Whitebox | APGD-CE | $95.8 \pm 1.9$ | $97.5 \pm 0.7$ |
| | APGD-DLR | $83.9 \pm 5.4$ | $90.7 \pm 2.1$ |
| | FAB | $5.4 \pm 2.2$ | $10.4 \pm 1.7$ |
| Blackbox-query | Square | $60.9 \pm 10.3$ | $21.9 \pm 2.8$ |
| | $\mathcal{N}$-Attack | $9.5 \pm 3.2$ | $56.7 \pm 8.9$ |
| Non-Interactive Blackbox | MI-Attack | $93.7 \pm 1.1$ | **99.9** $\pm 0.1$ |
| | DI-Attack | $95.9 \pm 1.6$ | **99.9** $\pm$ **0.1** |
| | TID-Attack | $92.8 \pm 2.7$ | $19.7 \pm 1.5$ |
| | SGM-Attack | N/A | **99.8** $\pm 0.3$ |
| | AEG (Ours) | **98.9** $\pm$ **1.4** | $98.5 \pm 0.6$ |

Table 4: Test error rates for average blackbox transfer over architectures at $\epsilon = 0.3$ for MNIST and $\epsilon = 0.03125$ for CIFAR-10 (higher is better)

| Source | Attack | VGG-16 | RN-18 | WR | DN-121 | Inc-V3 |
|---|---|---|---|---|---|---|
| | Clean | $11.2 \pm 0.9$ | $13.1 \pm 2.0$ | $6.8 \pm 0.7$ | $11.2 \pm 1.4$ | $9.9 \pm 1.3$ |
| | MI-Attack | $67.8 \pm 3.01$ | $86.0 \pm 1.7$ | **99.9** $\pm$ **0.1** | $89.0 \pm 2.6$ | $88.2 \pm 1.4$ |
| | DI-Attack | $68.3 \pm 2.4$ | $88.5 \pm 2.1$ | **99.9** $\pm$ **0.1** | $91.2 \pm 1.6$ | **91.5** $\pm$ **1.8** |
| WR | TID-Attack | $23.1 \pm 1.8$ | $25.9 \pm 1.3$ | $20.6 \pm 1.0$ | $23.6 \pm 1.2$ | $21.9 \pm 1.7$ |
| | SGM-Attack | $69.1 \pm 2.1$ | $88.6 \pm 2.0$ | **99.6** $\pm$ **0.4** | $90.7 \pm 1.9$ | $86.8 \pm 2.2$ |
| | AEG (Ours) | $40.8 \pm 3.22$ | $70.6 \pm 4.9$ | $98.5 \pm 0.6$ | **88.2** $\pm$ **4.6** | **89.6** $\pm$ **1.8** |
| | AEG (New) | **86.2** $\pm$ **1.8** | **94.1** $\pm$ **1.5** | $81.1 \pm 1.1$ | **93.1** $\pm$ **1.7** | $89.2 \pm 1.8$ |

Table 5: Error rates on $\mathcal{D}$ for average NoBox architecture transfer attacks with $\epsilon = 0.03125$ with Wide-ResNet architecture

## C.2 Qualitative Results

As a sanity check we also provide some qualitative results about the generated adversarial attacks. In Figure 3 we show the 256 attacked samples generated by our method on MNIST. In Figure 4 we show on the left the 256 CIFAR samples to attack, and on the right the perturbations generated by our method amplified by a factor 10.

## D    Implementation Details

We now provide additional details on training the representative classifiers and generators used in the AEG framework as outlined in 2. We solve the game using the ExtraAdam optimizer [29] with a learning rate of $10^{-3}$. We allow the generator to update its parameters several times on the same batch of examples before updating the critic. In particular we update the generator until it is able to fool the critic or it reaches some fixed number of iterations. We set this max number of iterations to 20 in all our experiments. We also find that biasing the critic update various forms of adversarial

Figure 3: Attacks generated on MNIST by our method.

| $d$-**ResBlock** |
|---|
| *Input*:$x$ |
| *Forward for computing* $F(x)$: |
| Reflection pad (1) |
| conv. (ker: $3\times3$, $d \to d$; stride: 1; pad: 1) |
| Batch Normalization |
| ReLU |
| Reflection pad (1) |
| conv. (ker: $3\times3$, $d \to d$; stride: 1; pad: 1) |
| Batch Normalization |
| *Output*:$x + F(x)$ |

Table 6: ResNet blocks used for the ResNet architectures (see Table 7) for the Generator. Each ResNet block contains skip connection (bypass), and a sequence of convolutional layers, normalization, and the ReLU non–linearity.

training consistently leads to the most effective attack. We reconcile this phenomenon by noting that through adversarial training the critic itself becomes a robust model which provides a richer learning signal to the generator. Furthermore, the elegance of the AEG frameworks allows the practitioner to further bias the optimization process of the critic —and consequently the generator— through picking and choosing effective robustness techniques such as training with PGD adversarial examples generated at a prior timestep.

## D.1 Generator Architecture

The architecture we used for the encoder and the decoder is described in Table 7 and 8. For MNIST we used a standard convolutional architecture and for CIFAR-10 we used a ResNet architecture.

Figure 4: **Left:** CIFAR examples to attack. **Right:** Pertubations generated by our method amplified by a factor 10. An interesting observation is that the generator learns not to attack the pixel where the background is white.

| Encoder |
| :---: |
| *Input: $x \in \mathbb{R}^{3 \times 32 \times 32}$* |
| Reflection Padding (3) |
| conv. (ker: $7 \times 7$, $32 \to 63$; stride: 1; pad: 0) |
| Batch Normalization |
| ReLU |
| conv. (ker: $3 \times 3$, $63 \to 127$; stride: 2; pad: 0) |
| Batch Normalization |
| ReLU |
| conv. (ker: $3 \times 3$, $127 \to 255$; stride: 2; pad: 0) |
| Batch Normalization |
| ReLU |
| 255-ResBlock |
| 255-ResBlock |
| 255-ResBlock |

| Decoder |
| :---: |
| *Input: $(\psi(x), z, y) \in \mathbb{R}^{256 \times 8 \times 8}$* |
| 256-ResBlock |
| 256-ResBlock |
| 256-ResBlock |
| Transp. conv. (ker: $3 \times 3$, $256 \to 128$; stride: 2; pad: 0) |
| Batch Normalization |
| ReLU |
| Transp. conv. (ker: $3 \times 3$, $128 \to 64$; stride: 2; pad: 0) |
| Batch Normalization |
| ReLU |
| ReflectionPadding(3) |
| conv. (ker: $7 \times 7$, $64 \to 32$; stride: 1; pad: 0) |
| Tanh |

Table 7: Encoder and Decoder for the convolutional generator used for the MNIST dataset.

### D.2 Baseline Implementation Details

The principal baselines used in the main paper include the Momentum-Iterative Attack (MI-Attack) [24], the Input Diversity (DI-Attack) [73], the Translation-Invariant (TID-Attack) [25] and the Skip Gradient Method (SGM-Attack) [71]. As Input Diversity and Translation invariant are approaches that generally can be combined with existing attack strategies we choose to use the powerful Momentum-Iterative attack as our base attack. Thus the DI-Attack consists of random input transformations when using an MI-Attack adversary while the TID-attack further adds a convolutional kernel ontop of the DI-Attack. We base our implementions using the AdverTorch [22] library and adapt all baselines to this framework using original implementations where available. In particular, when possible we reused open source code in the Pytorch library [59] otherwise we re-implement existing algorithms. We also inherit most hyperparameters settings when reporting baseline results except for number steps used in iterated attacks. We find that most iterated attacks benefit from additional optimization steps when attacking MNIST and CIFAR-10 classifiers. Specifically, we allot a 100 step budget for all iterated attacks which is often a five to ten fold increase than the reported setting in all baselines.

| | Decoder |
| --- | --- |
| | *Input:* $(\psi(x), z, y) \in \mathbb{R}^{64 \times 2 \times 2}$ |
| | Transp. conv. (ker: 3×3, 64 → 32; stride: 2; pad: 1) |
| **Encoder** | LeakyReLU(0.2) |
| | Max Pooling stride 2 |
| *Input:* $x \in \mathbb{R}^{28 \times 28}$ | Reflection Padding (3) |
| conv. (ker: 3×3, 1 → 64; stride: 3; pad: 1) | Transp. conv. (ker: 5×5, 32 → 16; stride: 3; pad: 1) |
| LeakyReLU(0.2) | LeakyReLU(0.2) |
| Max Pooling (stride: 2) | Max Pooling stride 2 |
| conv. (ker: 3×3, 64 → 32; stride: 2; pad: 1) | Transp. conv. (ker: 2×2, 16 → 1; stride: 2; pad: 1) |
| LeakyReLU(0.2) | Tanh |
| Max Pooling (stride: 2) | |

Table 8: Encoder and Decoder for the ResNet generator used for the MNIST dataset.

### D.3 Ensemble Adversarial Training Architectures

We ensemble adversarially train our models in accordance with the training protocol outlined in [67]. For MNIST models we train a standard model for 6 epochs, and an ensemble adversarial model using adversarial examples from the remaining three architectures for 12 epochs. The specific architectures for Models A-D are provided in Table. 8. Similarly, for CIFAR-10 we train both the standard model and ensemble adversarial models for 50 epochs. For computationally efficiency we randomly sample two out of three held out architectures when ensemble adversarially training the source model.

| A | B | C | D |
| --- | --- | --- | --- |
| Conv(64, 5, 5) + Relu | Dropout(0.2) | Conv(128, 3, 3) + Tanh | FC(300) + Relu |
| Conv(64, 5, 5) + Relu | Conv(64, 8, 8) + Relu | MaxPool(2,2) | Dropout(0.5) |
| Dropout(0.25) | Conv(128, 6, 6) + Relu | Conv(64, 3, 3) + Tanh | FC(300) + Relu |
| FC(128) + Relu | Conv(128, 6, 6) + Relu | MaxPool(2,2) | Dropout(0.5) |
| Dropout(0.5) | Dropout(0.5) | FC(128) + Relu | FC(300) + Relu |
| FC + Softmax | FC + Softmax | FC + Softmax | Dropout(0.5) |
| | | | FC(300) + Relu |
| | | | Dropout(0.5) |
| | | | FC + Softmax |

Table 9: MNIST Ensemble Adversarial Training Architectures)

## E   Further Related Work

Adversarial attacks can be classified under different threat models, which impose different access and resource restrictions on the attacker [1]. The whitebox setting, where the attacker has full access to the model parameters and outputs, thus allowing the attacker to utilize gradients based methods to solve a constrained optimization procedure. This setting is more permissive than the semi-whtiebox and the blackbox setting, the latter of which the attacker has only access to the prediction [57, 58] or sometimes the predicted confidence [36]. In this paper, we focus on a challenging variant of the conventional blackbox threat model which we call the NoBox setting which further restricts the attacker by *not* allowing any query from the target model. While there exists a vast literature of adversarial attacks, we focus on ones that are most related to our setting and direct the interested reader to comprehensive surveys for adversarial attacks and blackbox adversarial attacks [6, 16].

**Whitebox Attacks**. The most common threat model for whitebox adversarial examples are $l_p$-norm attacks, where $p \in \{2, \infty\}$ is the choice of norm ball used to define the attack budget. One of the earliest gradient based attacks is the Fast Gradient Sign Method (FGSM) [33], which computes bounded perturbations in a single step by computing the signed gradient of the loss function with respect to a clean input. More powerful adversaries can be computed using multi-step attacks such as DeepFool [55] which iteratively finds the minimum distance over perturbation direction needed to cross a decision boundary. For constrained optimization problems the Carlini-Wagner (CW) attack [14] is a powerful iterative optimization scheme which introduces an attack objective designed to maximize the distance between the target class and the most likely adversarial class. Similarly,

projected gradient descent based attacks has been shown to be the strongest class of adversaries for $l_2$ and $l_\infty$ norm attacks [53] and even provides a natural way of robustifying models through adversarial training. Extensions of PGD that fix failures due to suboptimal step size and problems of the objective function include AutoPGD-CE (APGD-CE) and AutoPGD-DLR (APGD-DLR) and leads to the state of the art whitebox attack in AutoAttack [20] which ensembles two other strong diverse and parameter free attacks.

**Blackbox Attacks**. Like whitebox attacks the adversarial goal for a blackbox attacker remains the same with the most common threat model also being $l_p$ norm attacks. Unlike, whitebox attacks the adversarial capabilities of the attacker is severely restricted rendering exact gradient computation impossible. In lieu of exact gradients, early blackbox attacks generated adversarial examples on surrogate models in combination with queries to the target model [57]. When given a query budget gradient estimation is an attractive approach with notable approaches utilizing black box optimization schemes such as Finite Differences [18], Natural Evolutionary Strategies [43, 46], learned priors in a bandit optimization framework [44], meta-learning attack patterns [26], and query efficient.

**Defenses**. In order to protect against the security risk posed by adversarial examples there have been many proposed defense strategies. Here we provide a non-exhaustive list of such methods. Broadly speaking, most defense approaches can be categorized into either robust optimization techniques, gradient obfuscation methods, or adversarial example detection algorithms [74]. Robust optimization techniques aim to improve the robustness of a classifier by learning model parameters by incorporating adversarial examples from a given attack into the training process [53, 67, 23]. On the other hand obfuscation methods rely on masking the input gradient needed by an attacker to construct adversarial examples [63, 12, 35, 21].

In adversarial example detection schemes the defender seeks to sanitize the inputs to the target model by rejecting any it deems adversarial. Often this involves training auxiliary classifiers or differences in statistics between adversarial examples and clean data [34, 54, 31].