[Reviews · NeurIPS 2020]

Review 1

Summary and Contributions: This paper tackles the non-interactive black-box attacks where attackers generate adversarial examples without any access to the target model. Based on theoretical analysis of this min-max two-player problem, it proposes a framework called "adversarial example games (AEG)" to draft transferable attacks to the entire function classes.

Strengths: 1. Compared with white-box or queryable black-box settings, black-box non-interactive setting is more challenging and less studied. Works to investigate adversarial attacks under this setting would benefit this community. 2. This paper formulates the transferable attack to "a certain hypothesis class of classifiers" as a zero-sum game between the adversarial generator and the classification model, which is new. It shows that the optimal adversarial distribution is the one that maximize a form of restricted conditional entropy over the target dataset.

Weaknesses: 1. In Appendix A, the proof of Proposition 1 is questionable, the Theorem 2 of "K. Fan. Minimax theorems" [26] says the min-max problem is equivalent to the max-min problem when the two-variable function is convex-concave. In the definition of $\Phi(f, g)$ in Eq. (AEG), it requires $\Phi$ to be convex on $f$ and concave on $g$. However, this is not satisfied when $f$ is the logit function defined by a deep nonlinear neural network. If Proposition 1 is not satisfied in the general case, then the existence of Nash Equilibrium cannot be claimed. 2. The gap, or the relationship, between the theory and the practice should be analyzed. There is a big jump from Section 3 to Section 4. How the architecture in Fig. 4 solves the AEG problem and how the optimizer (like Extra-Adam used here) is able to find the equilibrium should be briefly discussed in the mainbody of the paper. 3. The experimental results are weak. I cannot see the advantages of the proposed AEG over the baselines in all three parts. It is natural to ask this question: why should we use AEG instead of the traditional transfer attacks? >>> Post-rebuttal Regarding the Proposition 1, I agree with the authors in the rebuttal that \Phi is convex-concave w.r.t. **f, p_g**. However, the statement of proposition 1 uses g instead of p_g, as g is a function while p_g is a probability measure. I think function \Phi is not concave w.r.t. g, because the mapping from g to p_g is not linear. The validity of proposition 1 is still questionable. Regarding the experiments, the authors indeed answer some of my questions. But I am still concerned that AEG does not perform better than baselines in considerable cases. I am not saying AEG has no benefit in any setting, but I don't think AEG is better than traditional transfer attacks.

Correctness: I have a question regarding the Proposition 1, please see the previous section. The empirical methodology part is correct in general. In line 246, the authors say it is beneficial to include a pre-trained and fixed $f_s$ as a mechanism to incorporate prior knowledge during AEG optimization. However, I cannot find more information regarding this. There is no ablation study on this, or there is no more information about the settings, like architectures, of $f_s$. Minor: The adversarial budget $\epsilon$ used for CIFAR10 in the repo https://github.com/MadryLab/cifar10_challenge is 8/255, which is used in most literatures. The adversarial budget in this paper is slightly different, as 0.03125 = 8/256.

Clarity: This paper is well written in general. There is a big gap between the theoretical section and the experimental section. I think the high-level architecture used in the experiments, like Fig.4, should be included in the main text.

Relation to Prior Work: The previous related works are discussed comprehensively in this paper. It is clear.

Reproducibility: Yes

Additional Feedback: Due to the comments, questions and concerns above, I don't think this paper is ready to get published on NeurIPS and my overall score is, unfortunately, leaning towards a reject. However, I am very interested in the technical part of this work and happy to discuss with the authors. I welcome authors to clear my concern or point out some incorrect statements I made if any. I will re-evaluate the work after the rebuttal period. >>> Post-rebuttal Due to the concerns indicated in the weakness post-rebuttal section, I keep my evaluation the same.


Review 2

Summary and Contributions: This paper proposes adversarial example games, which targets a No-Box setting, where an attacker is trained without any access to the model, even on the query level. The adversarial example games is proposed in this setting, where the attacker try the maximize the power of attack on every possible model in the hypothesis set.

Strengths: 1. The No-Box attack setting is very novel. 2. The adversarial example game definition is sound. 3. On the simple setup of logistic regression with 2 classes, The paper obtains nice solution of the proposed optimization problem.

Weaknesses: 1. While the No-Box idea is very interesting, this attack scenario supposes the attacker can access a dataset from the data distribution, which is the same distribution from the training dataset. This requirement is still strong. For a well-generalized model, for example a CNN model on MNIST can often get 99% accuracy on the test dataset, it's very similar as having the model.. 2. The convergence of the optimization process is unclear. The paper claim the game will converge to an optimal attacker and a robust classifier. However, for a nonconvex model (i.e. DNN) that has a nonconvex loss, there's no guarantee that the optimization process will end well. 3. The experiment result doesn't show the AEG framework achieves much better result. The baselines are designed for white/black-box setting and then adapted to the No-box setting, and some of the attacks gets better result on more models. Also the method performs bad on transfering from the WR model. While the authors argue it's a special case on VGG-16, the performance on RN-18 is also not good, which shares a similar structure to WR. 4. The method is expensive. Combining these points, the significance of the proposed method is unclear. While the design is very nice, the optimization of AEG doesn't seem to work very well on real complicated models. --------------------------------------Post review--------------------------------------- Generally the authors provide feedback to my concerns. I don't feel my concerns are completely answered though: 1. For the answer "The NoBox attack demands the training set", the authors argue NoBox attack has a similar level of requirement as previous black-box attack. 2. Since the convergence of the method is unsure, there is no evidence the attack works better than some greedy approach, for example, train several models and generate some adversarial sample among all of them. 3. AEG is expensive. Combining these points, I feel while AEG has nice ideas, it's unnecessarily complicated for the current problem.

Correctness: Yes.

Clarity: Yes.

Relation to Prior Work: Yes.

Reproducibility: Yes

Additional Feedback: I believe this paper has a wrong subject area. It should be in "Social Aspects of Machine Learning -> Privacy, Anonymity, and Security".


Review 3

Summary and Contributions: The authors introduce the problem of a NoBox attack. This is meant as an extension of the black box attack where not even access to input-output pairs a given. Instead only the training and test dataset is provided. The main idea is to use a latent variable to train an adversarial generator and to use a large class of different architectures to train this generator.

Strengths: Instead of training one proxy network and computing universal adversarial examples with respect to this network a distribution of network architectures is used.

Weaknesses: * Proposition 1 seems to be false. In order to replace the minmax with a maxmin problem it is not enough to show that the domain over which \phi is minimized is convex. \phi itself has to be a convex function on this convex domain. It is neither clear nor proven why this should be the case. * Section 4.1 only studies linear classifiers. Therefore, Proposition 2 is only true in this context. This is not made sufficiently clear in the phrasing of Proposition 2. * Q3 of Section 5 claims to attack robust classifiers. Nonetheless no provably robust classifier is attacked

Correctness: See above.

Clarity: The sentences are well structured.

Relation to Prior Work: I am not missing any relevant paper.

Reproducibility: Yes

Additional Feedback: Post-rebuttal comments: My main concerns were with respect to Proposition 1 and 2. The rebuttal did not address these concerns properly. If X is a random variable, g(X) is a different random variable with its distribution p_g. It is not clear why the authors identify p_g with g in their rebuttal. Overall, the presented math is below the quality that people would expect from NeurIPS papers. Therefore, I recommend to reject the paper.


Review 4

Summary and Contributions: This work proposed NoBox, short for non-interactive BlackBox attack. It aims to attack a known function family F at its entirety, which differs from the conventional adversarial attack focusing on testing instances and specific architectures. The mathematical justification and experiments are both presented.

Strengths: This paper does have a novel perspective: attacking a function space at its whole. The mathematical proof looks sound to me. At the nash equilibrium, the attack is theoretically effective to any function from a given class. With that being said, the threat model is interesting and may open a new venue of research.

Weaknesses: I have a few concerns regarding this paper. 1. How practical/realistic is the threat model? In the paper, chapter 2.2, the NoBox attack demands the training set. In the real world however, isn't the training set even more precious than the trained models? 2. In the AEG objective, the generator needs to get the gradient to be trained. Would AEG still applicable to the non-differentiable robust classifiers, such as: [1] Countering Adversarial Images using Input Transformations [2] THERMOMETER ENCODING: ONE HOT WAY TO RESIST ADVERSARIAL EXAMPLES [3] Retrieval-Augmented Convolutional Neural Networks against Adversarial Examples The common point of these approaches is they all incorporate some sort of in-differentiability. 3. The experiments. (Maybe my misunderstanding) Many published papers in this field used ImageNet (and the top-1 score) to benchmark the effectiveness of the attack or the robustness of the defense. However this paper the experiments are limited to only CIFAR and MNIST. 4. One experiment I'd like to request: (i)- get a model trained on some dataset at epoch N, N+1, N+2... N+k (ii)- use the generator to generate pertubed imagess to attack all of them. (iii)- show the effectiveness of the attack. This should be a more realistic scenario and it aligns with the main point. 5. A portion of the experiment has compared the NoBox attack to the other attacks. These are generally under different threat model assumptions. However the main claim of the paper is that the NoBox is capable of attacking different models in the same function space. It would be better if the authors can present the NoBox's effectiveness attacking more diversified trained neural network models.

Correctness: Yes

Clarity: Yes

Relation to Prior Work: Yes

Reproducibility: Yes

Additional Feedback:

[Author Response · NeurIPS 2020]

We thank the reviewers for their feedback and helpful comments, however there are important misunderstandings which we now clarify and politely ask for another evaluation using this rebuttal as context. We appreciate that **R1**, **R2** and **R5** acknowledged the interest, novelty and challenges of the No-Box threat model as well as the soundness of the Adversarial Example Games formulation. We believe these two contributions —i.e., proposing a novel and more realistic threat model and at the same time providing a framework to study and solve it—are extremely relevant to a venue that cares for new research perspectives. We are also encouraged that **R1**, **R2**, **R4** and **R5** felt that our submission was well written and the main thrust of the paper was clear. Finally, we are pleased by the interest of **R1** and **R4** in the proofs provided in the appendix. We start by clarifying their concern about Prop. 1.

**Concern regarding Proposition 1 (R1, R4):.** The concern regarding the absence of convex-concavity of the objective function. As mentioned L550, we need the loss function $\ell$ to be convex (this is the case for any standard loss function such as the cross entropy-loss or the mean-square loss). In (12), since $(x', y) = (g(x, y, z), y) \sim p_g$ we have,

$$\varphi(f, g) = \varphi(f, p_g) = \mathbb{E}_{(x', y) \sim p_g}[\ell(f(x'), y)]. \tag{1}$$

Then, because the loss function $\ell$ is convex for any $g$, we have $f \mapsto \varphi(f, g)$ that is convex. Regarding the concavity of the payoff with respect to the second variable, by linearity of the expectation with respect to $p_g$ the function $p_g \mapsto \varphi(f, p_g)$ is linear and thus concave. Overall the payoff $(f, p_g) \mapsto \varphi(f, p_g)$ is convex concave and the set to which belongs $p_g$ is convex and compact (c.f. L566 and L569). Thus, we can apply Fan's Theorem (Theorem 1 L573).

**The NoBox attack demands the training set (R2, R5):.** We appreciate the concern echoed by **R2**, **R5** regarding the ability of the adversary to sample a training set from the same distribution as the target model as a strong requirement. We disagree. We argue that every single blackbox transfer attack paper, including all baselines we use, requires an available training set in order to first train a source model which can then be used to learn transferable attacks. Indeed, without access to such a training set it would not be possible to train a source model thus preventing possible attacks. Also, we note that Defn 3. in the seminal work by Tramèr et al., [2018] (ICLR 2018) standardizes this requirement.

**Experimental setup and results (R1, R2, R5):.** We value the feedback shared by **R1** and **R2** regarding the utility of AEG over other blackbox transfer attack strategies. Our work is the first to propose a principled way to craft transferable adversarial examples to function classes while all other prior transfer strategies, while powerful, are heuristically motivated. In terms of raw performance, AEG is SOTA or within $1\%$ of the best transfer method for known architecture attacks, which is the precisely the setting our theory applies to. When transferring to different architectures and possibly differing function classes AEG is still SOTA when using a RN-18 or DN-121 as the source model. For robust models, we expect a drop in performance for ensemble adversarial training as the theory suggests our representative classifier cannot cope with a set union of multiple function classes. While for PGD-Adversarial attacks we are again SOTA for all transfer attacks highlighting how our generator is optimized against a worst case adversary. Thus we would like to politely push back against the assertion by **R1** that there is no benefit to AEG in any setting. Regarding **R5**'s concern on attacking non-differentiable robust models, the elegance of our framework gives an affirmative answer as long as this robust model is within the same function class. In practice this amounts to training with a non-differentiable $f_c$. We would also like to gently clarify to **R1** that the $\epsilon$-budget in our CIFAR10 experiments is identical to the Madry challenge as the range for pixel values $0 - 255$ is inclusive of zero.

**How can we converge in practice (R1, R2):.** The convergence of the optimization process in a game with general non-convex losses is an *open question in the field* (see Lin, Jin & Jordan [2019] or Kodali et al. [2017] for discussions). Similarly as the original GAN paper [Goodfellow et al, 2014] and any practical GAN paper, proving the convergence of the gradient based optimization algorithm is beyond the scope of this paper. However, note that we use an extrapolation based method (Extra-Adam) as extrapolation is a principled method for minimax optimization proposed by Gidel et al. [2019] that, unlike Adam, has convergence guarantees in the convex-concave setting.

**How the architecture solves the problem (R1):.** The architecture Fig. 4 corresponds exactly to the theory. The generator takes as input $x, y$ and a latent variable $z$ (that depends on $x$ in order to exploit the structure of the input and outputs $x <= x + g(x, y, z)$ such that $\|g(x, y, z)\|_\infty \leq \epsilon$. The critic $f$ takes as input $x'$ and $y$ and occurs a loss $\ell(f(x'), y)$. That loss is exaclty the one described in (AEG). We will further clarify these points in our submission.

**The AEG method is expensive (R2):.** We agree that AEG is expensive to train. But, in the context of adversarial attack what matters more is the computational cost of crafting an adversarial example at test time (the attacker may pay a cost for taking too much time to craft an adversarial example). In that context, our method is by far among the fastest since it only requires the inference of the generator network while other standard methods require to solve an optimization problem by querying many gradients (or function values) of a predictor function.

**Q3 of Section 5 claims to attack robust classifiers (R4).** We acknowledge **R4** comment regarding our method against robust classifiers. While it is true that these approaches are not provably robust, to the best of our knowledge no such method exists to certify deep architectures such as Wide Resnet on MNIST and CIFAR10. Furthermore, both the defense methods studied in our work are the defacto gold standards for transferable adversarial examples thus we feel our investigation against robust models in the NoBox setting is appropriate.

[Meta-Review · NeurIPS 2020]

The paper develops some interesting game-theoretic results in the "No Box" setting, in which neither the attacker nor the defender have access to other party's actions. It proves that some standard attack and defense strategies constitute a Nash equilibrium in this setting. However, on the practical part, the experimental evaluation does not always demonstrate the utility of the proposed method, especially in comparison to the baselines. It would also be helpful if the authors could discuss the relationship of their method to prior game-theoretic approaches to adversarial learning in the general machine learning setting, e.g. Brückner, M., Kanzow, C. and Scheffer, T., 2012. Static prediction games for adversarial learning problems. The Journal of Machine Learning Research, 13(1), pp.2617-2654. Brückner, M. and Scheffer, T., 2011, August. Stackelberg games for adversarial prediction problems. In Proceedings of the 17th ACM SIGKDD international conference on Knowledge discovery and data mining (pp. 547-555).